# The Preparation and Properties of Terephthalyl-Alcohol-Modified Phenolic Foam with High Heat Aging Resistance

**DOI:** 10.3390/polym11081267

**Published:** 2019-07-31

**Authors:** Tiejun Ge, Xiaoqi Hu, Kaihong Tang, Dongqi Wang

**Affiliations:** 1Plastic Engineering Research Center of Shenyang University of Chemical Technology, Shenyang 110142, China; 2Liaoning Polymer Materials Engineering and Technology Research Center, Shenyang 110142, China; 3Shenyang Huada and Kangping Plastic Woven Research Institute, Shenyang 110142, China

**Keywords:** phenolic modification, phenolic foam, heat aging resistance, terephthalyl alcohol

## Abstract

In this experiment, terephthalyl alcohol was used as a modifier to modify phenol under both acidic and alkaline conditions to obtain modified phenols with different molecular structures. Subsequently, the modified phenols reacted with paraformaldehyde in an alkaline environment. After foaming and curing, a modified phenolic foam with high heat aging resistance was obtained. The molecular structure was characterized via Fourier transform infrared spectrometry (FT-IR) and nuclear magnetic resonance spectroscopy (^13^C NMR). The results showed that two different structures of phenolic resin can be successfully prepared under different conditions of acid and alkali. The modified phenolic foam was tested by thermogravimetric analysis. In addition, the modified phenolic foam was tested for mass change rate, dimensional change rate, powdering rate, water absorption rate, and compressive strength before and after aging. The results show that the modified phenolic foam has excellent performance. After heat aging for 24 h, the mass loss rate of the modified phenolic foam obtained by acid catalysis was as low as 4.5%, the pulverization rate was only increased by 3.2%, and the water absorption of the modified phenolic foam increased by 0.77%, which is one-third that of the phenolic foam. Compared with the phenolic foam, the modified phenolic foam shows good heat aging resistance.

## 1. Introduction

Known as the “king of insulation materials”, phenolic foam is a kind of foam material with good heat insulation, good sound insulation, low thermal conductivity, and an easy molding process [1,2]. In addition, phenolic foam also has the advantages of resistance to flame penetration, an absence of dripping, a self-extinguishing ability, and low smoke generation in the event of an open flame [3,4]. Phenolic resin as a foaming matrix is a resole phenolic resin obtained by polycondensation of phenol and paraformaldehyde. In the synthesis process, paraformaldehyde is first reacted with an active point on the phenol to form hydroxymethylphenol [5,6]. The hydroxymethylphenol then continues to react with the active sites on the phenol (active hydrogen in the ortho and para position) to form a methylene bridge [7,8]. However, this structure in which only the methylene group is bonded to the benzene ring results in a large brittleness of the resin [9]. In addition, the presence of phenolic hydroxyl groups in the resin increases the water absorption of the resin, and the phenolic hydroxyl groups are not heat-resistant. Many research scholars have studied the modification of phenolic resins. For example, Yajun Guo et al. used nanoSiO_2_ to modify phenolic resin in order to enhance thermal stability [10]. Shimin Kang et al. used Humins to synthesize Humin-Phenol-Formaldehyde [11]. Yu Hu et al. used epoxy- and methacrylate-functionalized silica sols-modified phenolic resin to improve the ablation resistance of their glass fiber-reinforced composites [12]. Zeya Li et al. synthesized iron-modified phenolic resin by grafting iron ions into a phenolic resin structure via a coordination reaction [13]. Shengyong You et al. synthesized KH560-modified phenolic resin by chemical synthesis using the silane coupling agent, KH560 [14]. Disadvantages associated with phenolic resins adversely affect the performance of phenolic foams. Therefore, improving the performance of the foamable phenolic resin and modifying the phenolic resin in chemical structure is a means to improve the performance of phenolic foam [15,16,17]. Phosphate-containing cardanol-modified phenolic resin, prepared by Caiying Bo et al., can overcome the brittleness of the foam, improve its flame retardancy, and attain a high thermal stability and good mechanical properties [18]. In addition, the phenolic resin may be physically modified [19,20]; that is, the modifier is blended with the phenolic resin and then foamed as a matrix resin. Yufeng Ma et al. added an environmentally friendly, halogen-free flame retardant, a synergist, and other additives to the phenolic resin and foam, and the obtained modified foam had excellent flame retardancy [21]. Xingming Hu et al. added polyethylene glycol (PEG) with different molecular weights and different parts to phenol-urea-formaldehyde foam. The results showed that polyethylene glycol significantly improved the toughness and impact strength of the foam, but the thermal stability of the foam had a slightly adverse effect [4].

It can be seen that ordinary phenolic foam cannot meet the increasingly stringent requirements of high-performance foams in the current market. Therefore, improving the performance of the foamable phenolic resin and improving the performance of various aspects of the phenolic foam is the current research direction of phenolic foam insulation materials [22,23,24,25,26]. Our research team uses a variety of modifiers to modify phenolic resin. We test and study the properties of the modified phenolic foam. For example, we replaced a portion of phenol with m-pentadecylphenol to introduce a fifteen-carbon long chain into the phenolic resin structure [27]. After the acetyl acetate-terminated polyether was synthesized, it was introduced into the phenolic resin [28]. Copolymerization of polyethylene adipate glycol (PEA) was achieved during the synthesis of phenolic resin [29]. The foam properties, especially the toughness, of the modified phenolic resin as a foaming matrix were greatly improved. In addition to modifying the structure of the phenolic resin during the synthesis of the phenolic resin, our research team also physically modified the phenolic resin. For example, nano-silica, nano-alumina, hollow glass microbeads [30], and different amounts of urea were used to modify phenolic resin [31], and the modified resin was prepared as a foam for testing. The results showed that the toughness and water absorption of the foam were improved. Although we have done much research on the toughness of phenolic foam, few research experiments have improved its aging resistance. Therefore, this paper studies the modification of the thermal aging properties of phenolic foam.

Terephthalyl alcohol is one of the raw materials of xylok resin [32,33,34]. An aralkyl group was introduced into the resin skeleton by a chemical reaction, thereby imparting excellent heat resistance, water absorption resistance to the material and improving the dielectric properties of the material. In order to improve the thermal stability of phenolic foam [35] as well as its aging resistance, this experiment chose terephthalyl alcohol to chemically modify phenolic foam. Hui Liu et al. prepared a thermoset 4,4′-biphenol modified phenolic resin with a high residual carbon ratio and high thermal stability by using 4,4′-biphenol, phenol, alkylphenol, and formaldehyde [36]. In order to improve the thermomechanical properties and thermal stability, Zixuan Lei used epoxy-polyhedral oligomeric silsesquioxane-modified novolac phenolic resin, and then prepared a novel organic–inorganic hybrid network with hexamethylenetetramine [37]. The phenol structure contains an active site to which terephthalyl alcohol can react with and attach the aralkyl group to the phenol structure. After the modified phenol with a different molecular structure is obtained, a modified phenolic foam with excellent heat aging resistance was prepared using paraformaldehyde.

## 2. Materials and Methods

### 2.1. Materials

Phenol, paraformaldehyde, p-toluenesulfonic acid (as a catalyst and hardener), sodium hydroxide (as a catalyst), and n-pentane (as a foaming agent) were purchased from the Tianjin Damao Chemical Reagent Factory (Tianjin, China). Terephthalyl alcohol was purchased from Zhengzhou Alpha Chemical Co., Ltd. (Zhengzhou, China). Tween-80 (as a surfactant) was purchased from Sinopharm Chemical Reagent Co., Ltd. (Shanghai, China). The raw materials mentioned above are all analytical reagent.

### 2.2. Preparation of Modified Phenolic Resin

The composition of the modified phenolic resin (acid catalyzed) is as follows: The amount of terephthalyl alcohol was 5–20% (step: 5%) of phenol consumption. The amount of p-toluenesulfonic acid added was 0.5% of the total amount of phenol. The molar ratio of phenol to paraformaldehyde was 1:1.5. The amount of sodium hydroxide used was the sum of the dosage of neutralizing p-toluenesulfonic acid and 1% of the mass of the phenol.

According to the formula, a certain amount of phenol solution was added to a three-necked flask (Tianjin Damao Chemical Reagent Factory, Tianjin, China) containing a stirrer, a thermometer, and a glass manifold, and the upper part was connected with a reflux condenser. Terephthalyl alcohol was added to the reaction system. After stirring uniformly, the catalyst p-toluenesulfonic acid was added at 70–75 °C and stirring was continued. Polycondensation was carried out for 1 h at 130 °C. The by-products were separated by the glass manifold and the reflux condenser. After the reaction system was cooled to 65 °C, sodium hydroxide was added. Paraformaldehyde was added to the three-necked flask in batches, and the mixture was heated for 1 h. Finally, the temperature was raised to 90 °C, and the reaction was continued for 30 min to obtain a modified phenol resin.

The composition of the modified phenolic resin (base catalyzed) is as follows: The amount of terephthalyl alcohol was 5–20% (step: 5%) of phenol consumption. The molar ratio of phenol to paraformaldehyde was 1:1.5, and the sodium hydroxide was 1% by mass of phenol.

According to the formula, certain amounts of phenol solution and sodium hydroxide were added to a three-necked flask (Tianjin Damao Chemical Reagent Factory, Tianjin, China) containing a stirrer, a thermometer, and a glass manifold, and the upper part was connected with a reflux condenser. After stirring uniformly, terephthalyl alcohol was added to the reaction system. Stirring continued. The polycondensation was carried out for 1 h at 130 °C. The by-products were separated by the glass manifold and the reflux condenser. The reaction system was then cooled to 65 °C, and the paraformaldehyde was added in batches for about 30 min, and the mixture was heated for 1 h. Finally, the temperature was raised to 90 °C, and the reaction continued for 2–2.5 h to obtain a modified phenolic resin.

### 2.3. Preparation of Modified Phenolic Resin Foam

The above modified phenolic resin was used for foaming. The modified phenolic resin foaming formula is shown in Table 1. The surfactant was Tween-80, the foaming agent was n-pentane, and the curing agent was p-toluenesulfonic acid. The surfactant and the foaming agent were sequentially added to the modified phenolic resin. After stirring quickly, the curing agent was added and stirring continued. At last, the mixture was uniformly mixed and poured into a mold, which was solidified in an oven (Beijing ever bright medical treatment instrument co., Ltd., Beijing, China) at 80 °C and taken out after about 10 min. Thus, a modified phenolic foam was obtained.

## 3. Characterization

### 3.1. Characterization of Resin

The modified phenolic resin was subjected to an FTIR test by Fourier transform infrared spectrometer (NEXUS 470 Thermo Electron Corporation, Shanghai, China), and the sample was evenly spread on a potassium bromide sheet and placed on an FTIR instrument for testing.

The structure of the modified phenolic resin was characterized by a nuclear magnetic resonance (NMR) spectrometer (AVANCE-III-500MHz, Bruker Biospin, Munich, Germany). Six to eight milligrams of terephthalyl-alcohol-modified phenol product was dissolved in a solvent deuterated dimethyl sulfoxide (DMSO). The main internal standard was tetramethyl. Silane (TMS) was characterized via nuclear magnetic resonance carbon spectrum (^13^C NMR) operating at 125.77 MHz.

The viscosity of the modified phenolic resin was tested according to Chinese National Standards (GB/T 14074.3-1993).

The water content of the modified phenolic resin was tested according to Chinese National Standards (GB/T 14074.11-1993).

The free phenol of the modified phenolic resin was tested according to Chinese National Standards (GB/T 14074.13-1993).

### 3.2. Characterization of Foam

The thermal properties of the foam were tested using a thermogravimetric (TG) analyzer (TG209F1 type, Germany NETZSCH company, Selb, Germany). The size of the foam was two-thirds of the alumina crucible. The test conditions were a nitrogen atmosphere, with a temperature range of 40–800 °C and a heating rate of 20 °C /min.

The dimensional change rate was tested according to Chinese National Standards (GB/T 8811-88). The foam was cut to a size of 100 × 100 × 25 mm. The length and width were measured at three different positions, and the thickness was measured at five different positions according to the standard, aging at a temperature of 70 °C. The foam was taken out every 12 h for measurement and then placed in an oven to continue aging. Finally, the average dimensional change rate to length, width, and thickness was calculated.

The pulverization rate was tested according to Chinese National Standards (GB/T12812-1991). The foam was cut to a size of 50 × 50 × 50 mm. Subsequently, 200 g of weight was placed on the foam, and the sample was pulled 20 times at the same distance horizontally on 46 μm sandpaper. The mass before and after was compared, and the pulverization rate was calculated.

The water absorption rate was tested according to Chinese National Standards (GB/T8810-2005). The foam before and after aging was immersed in distilled water for 24 h, and subsequently weighed and compared with the foam quality before immersion. The water absorption rate was then obtained.

The compressive strength was tested according to Chinese National Standards (GB/T 8813-2008) using an RGL-type microcomputer control electronic universal testing machine (Shenzhen Rui Geer Instrument Co., Ltd., Shenzhen, China). The foam was cut to a size of 100 × 100 × 50 mm, and the foam was compressed, at a certain rate, to 85% of the original thickness. The compressive strength was recorded.

## 4. Results and Discussion

### 4.1. Characterization of Modified Phenolic Resin

#### 4.1.1. Mechanism of Terephthalyl Alcohol-Modified Phenol

Figure 1a is an FTIR spectrum of phenol pre-modified with terephthalyl alcohol under acidic conditions, and Figure 1b is an FTIR spectrum of phenol pre-modified with terephthalyl alcohol under alkaline conditions. It can be seen in Figure 1a that there is a strong aromatic ether stretching vibration peak at 1000~1050 cm^−1^ at Location A, which indicates that phenol reacts with terephthalyl alcohol under acidic conditions to form an aromatic ether. It can be seen in Figure 1b that there is a strong stretching peak of -CH_2_- at 2923 cm^−1^ at Location a, which indicates that, under the alkaline condition, the reaction of terephthalyl alcohol with phenol forms a methylene bridge. The methylene group connects the phenol to the benzene ring of terephthalyl alcohol. Figure 1c is a ^13^C HMR spectrum of phenol modified with terephthalyl alcohol under acidic conditions. Figure 1d is a ^13^C HMR spectrum of phenol modified with terephthalyl alcohol under alkaline conditions. It can be seen in Figure 1c that, under acidic conditions, there are only four types of carbon atoms in the structure of the product of terephthalyl-alcohol-modified phenol, where A, B, and C represent the chemical shifts of the ortho, meta, and para positions of the benzene ring, respectively, and the chemical shift of carbon atom is substituted by a substituent on the benzene ring at Location D. It can be seen in Figure 1d that Location a is the chemical shift of the carbon atom substituted by the substituent on the phenol ring, and Location b is the chemical shift of the carbon in the methylene group connecting the phenol and the terephthalyl alcohol. It can be deduced from the infrared spectrum and the nuclear magnetic carbon spectrum that different structures of phenol are modified by terephthalyl alcohol under the conditions of two different catalysts. Under acidic conditions, terephthalyl alcohol reacts with phenol to form an aromatic ether, which connects the benzene ring to the phenol structure through an ether bond. Under alkaline conditions, the ortho and para positions of phenol are substituted by terephthalyl alcohol, and the benzene ring is attached to the phenol through a methylene bridge.

#### 4.1.2. Infrared Spectrum of Modified Phenolic Resin

The FTIR spectrum of phenolic resin and modified phenolic resin is shown in Figure 2. It can be deduced from the spectrum that the stretching vibration peak of the C=C skeleton of the benzene ring is at 1597.22 cm^−1^. The 824.79 cm^−1^ and 757.67 cm^−1^ correspond to the absorption peaks of the benzene ring para and ortho position substitution, respectively. At 2933.56 cm^−1^, there is a stretching vibration peak of methylene C–H. At 1013.36 cm^−1^, there is a stretching vibration peak of C–O in a methylol group. At 3330 cm^−1^, there is a stretching vibration peak of a hydroxyl group (–OH), and there is a bending vibration peak of a methylene group at 1463.35 cm^−1^. The weaker peak of the modified resin prepared under the acid condition at 1151 cm^−1^ is the vibration peak of the aromatic ether bond formed by the reaction of terephthalyl alcohol with the phenolic hydroxyl group under acidic conditions. The modified phenolic structure was obtained by analyzing the nuclear magnetic resonance spectrum and the infrared spectrum as shown below.

It can be deduced from the above that phenolic resin with a different molecular structure was prepared with phenol, terephthalyl alcohol, and paraformaldehyde under different conditions of acid and alkali. Two molecular structural fomulas are shown in Figure 3a,b. Under alkaline conditions, the ortho and para positions of phenol are substituted by terephthalyl alcohol, introducing a benzene ring with extremely high thermal stability into the resin structure, so the phenolic foam that uses modified resin as foaming material has good thermal stability. Under acidic conditions, terephthalyl alcohol not only introduces the benzene ring into the resin structure, but also blocks the active phenolic hydroxyl group by reacting with the hydroxyl group. The presence of the phenolic hydroxyl group affects the water absorption of the resin and renders the resin non-resistant to aging. Therefore, the consumption of a part of the phenolic hydroxyl group can produce foam material high in aging resistance.

Table 2 shows the basic properties of the modified phenolic resin, wherein the modifier is used in an amount of 15%. The viscosity of the resin used for foaming is generally controlled at 3 to 5 Pa·s, the water content is in the range of 6% to 8%, and the free phenol is controlled to be less than 5% [38]. From the table, it is known that the modified phenolic resin meets resin foaming conditions.

### 4.2. Modified Phenolic Foam Heat Resistance and Aging Resistance Analysis

#### 4.2.1. Thermogravimetric Analysis of Modified Phenolic Foam

Figure 4 and Table 3 show the thermogravimetric analysis and typical parameters of phenolic foam and terephthalyl-alcohol-modified phenolic foam (acid-catalyzed/base-catalyzed). As can be seen in the figure, in the initial stage of the thermal weight loss of phenolic foam, volatilization of small molecules in the foam including water molecules, free formaldehyde, and free phenol leads to a decline in quality. Modified phenolic foam reduces the free formaldehyde content and free phenol content in the resin due to the addition of terephthalyl alcohol. Therefore, the heat-resistant temperature of the modified phenolic foam was higher than that of the phenolic foam at a weight loss of 5%, especially under the modified phenolic foam prepared under acidic catalytic conditions, up to about 100 °C higher than phenolic foam. The maximum temperatures of SGO/PF and ZGO/PF, reported by Xiaoyan Li et al., at 5% weight loss are 199.1 °C [2] and 204 °C [35], respectively, while the temperature of 5% weight loss of terephthalyl-alcohol-modified phenolic foam was as high as 267.58 °C. The thermal decomposition peak was also improved compared to the foam. In the final stage of the thermal weight loss at 800 °C, the residual carbon ratio of the modified phenolic foam was still higher than that of the phenolic foam. This is because the introduction of terephthalyl alcohol increases the aryl content, so the residual carbon ratio increases. In summary, the modified phenolic foam has a higher heat resistance than the phenolic foam.

#### 4.2.2. Dimensional Change Rate of Modified Phenolic Foam under Thermal Action

Figure 5a–c shows the dimensional change rate in the length, width, and thickness of the phenolic foam and the modified phenolic foam (15% of the amount of terephthalyl alcohol). The foam was aged in the oven at 70 °C, and the size gradually became smaller. It can be seen in the figure that the dimensional change rate of phenolic foam increases with the increase in aging time, and the highest change rate reaches 5.3% at 120 h. However, the dimensional change rate of the modified phenolic foam was less than or equal to 4% within 120 h of aging. The modified phenolic foam has a smaller dimensional change rate than the phenolic foam under the same aging time. When the amount of terephthalyl alcohol added is the same, while the catalysts are different, the dimensional change rates are slightly different. This is due to the introduction of terephthalyl alcohol, which results in the existence of a thermally stable benzene ring structure in the molecular structure, forming a spatial network structure that is not easily destroyed under heat. Therefore, the modified phenolic foam has a smaller dimensional change rate than the phenolic foam under conditions of prolonged heating.

#### 4.2.3. Mass Loss Rate of Modified Phenolic Foam

Figure 6 is a graph showing the change in mass loss rate of phenolic foam and modified phenolic foam after aging for 24 h in an oven at 70 °C. It can be seen in the figure that the mass loss rate of the phenolic foam reached 8.2% after heat aging for 24 h in an oven at 70 °C. After adding the modifier terephthalyl alcohol, the mass loss rate of the modified phenolic foam decreased significantly, and the mass loss rate gradually decreased with the increase of the amount of terephthalyl alcohol. The minimum value was reached when the amount of terephthalyl alcohol added was 15% of phenol: 4.5% for the acid catalytic modification of phenolic foam and 4.9% for the base catalytic modification. When the addition amount was 20%, the mass loss rate of the modified phenolic foam increased but was still smaller than the mass loss rate of the phenolic foam. In summary, the introduction of terephthalyl alcohol reduces the mass loss rate after the thermal aging of phenolic foam. This indicates that terephthalyl alcohol will reduce the content of small molecules such as free aldehydes in the resin after structural modification of the phenolic resin, and the modified phenolic resin has a stable structure and will not decompose, further improving the thermal stability of the foam.

#### 4.2.4. Effect of Heat Aging on the Degree of Phenolic Foaming

Figure 7 shows the change in the pulverization rate of phenolic foam and modified phenolic foam before and after aging at 70 °C for 24 h. The phenolic resin after curing is a structure composed of a methylene-linked benzene ring structure, the resin is brittle, and the phenolic foam is easily pulverized. It can be seen in the figure that the pulverization rate of the modified phenolic foam is significantly reduced. This is because the addition of terephthalyl alcohol will form a new spatial network structure and form aromatic ethers under acidic modification conditions, changing the state such that the phenolic resin has only a methylene linkage, and the pulverization rate is significantly reduced. After heat aging at 70 ° C for 24 h, the pulverization rate of the phenolic foam increased by 4.38%, while the degree of pulverization of the modified phenolic foams under acidic modification conditions only increased by 3.2% and the degree of pulverization of the modified phenolic foams under alkaline modification conditions only increased by 3.9%. This further shows that the addition of terephthalyl alcohol introduces a highly stable benzene ring in the phenolic resin, which can affect the thermal aging of the foam and prevent the increase of the pulverization rate caused by the increase of the external temperature, which proves that the structure of the modified resin is stable, and the modified foam has an improved heat resistance relative to ordinary phenolic foam.

#### 4.2.5. Effect of Heat Aging on Water Absorption of Modified Phenolic Foam

Figure 8 is a schematic diagram showing the change of water absorption rate of phenolic foam and phenolic foam before and after aging at 70 °C for 24 h. It can be seen in the figure that the water absorption rate of the modified phenolic foam is significantly lower than that of the phenolic foam before aging. This is because terephthalyl alcohol is introduced into the phenolic resin structure by a chemical reaction, and the molecular weight of the system rises. During the foaming process, the volatilization of the blowing agent is limited, the cell expansion and dispersion are difficult, and a small and dense cell structure is formed after solidification, resulting in a decrease in the water absorption rate. In addition, the phenolic resin obtained under acid catalysis further reduces the water absorption rate of the foam due to the blocking of the phenolic hydroxyl group of phenol by terephthalyl alcohol. After heat aging, the structure of the foam is destroyed to a certain extent due to high temperature, the cells are opened, and the opening ratio is increased, resulting in an increase in the water absorption rate. The water absorption rate of the phenolic foam increased by 2.15%, while the modified foams under acidic modification conditions only increased by 0.77% and the modified foams under alkaline modification conditions only increased by 1.23%. This indicates that the terephthalyl-alcohol-modified phenolic resin introduces a benzene ring with extremely high thermal stability into the resin structure, which can prevent the foam from breaking under the aging effect and improve the heat aging resistance of the foam.

#### 4.2.6. Effect of Heat Aging on the Compressive Strength of Modified Phenolic Foam

Figure 9 shows the change of compressive strength with phenolic foam and modified phenolic foam at an aging temperature of 70 °C. It can be seen in the figure that the introduction of the modifier terephthalyl alcohol increases the compressive strength of the modified phenolic foam when not subjected to heat aging. This is because the benzene ring content of the modified phenolic resin is increased, the bond energy of the benzene ring is high, and the structure of the modified resin after foaming is dense and stable, such that the foam can withstand greater compressive strength. After 48 h of heat aging, the compressive strength of phenolic foam decreased slightly, the phenolic foam decreased by 47%, while the modified phenolic foam under acidic modification conditions only decreased by 20% and the modified phenolic foam under alkaline modification conditions only decreased by 25%. This indicates that the modified foam obtained by structurally modified phenolic resin by terephthalyl alcohol is more resistant to aging than the phenolic foam and has a higher thermal stability. The phenolic foam modified under acidic conditions is higher in heat aging resistance due to the presence of an ether bond in its molecular structure.

#### 4.2.7. Modified Phenolic Foam Heat Aging Color Change

Table 4 is a schematic diagram showing the apparent color change of the foam over time at 70 °C for phenolic foam and modified phenolic foam. It can be seen in the figure that the color of the foam gradually changes from light pink to reddish brown under heat and finally to purplish black. This is due to the formation of a quinone structure under the heat aging of the phenolic hydroxyl group in the resin structure. The color of the benzoquinone is red, so the color of the phenolic foam got darker. The phenolic foam has a darker color than the modified phenolic foam when it is not heated, and the color becomes purplish black when heated for 12 h, while the modified phenolic foam exhibits a purplish black color at 72 h. The addition of terephthalyl alcohol makes the foam aging rate slow, and the modified phenolic foam obtained under alkaline conditions is darker in color than the modified phenolic resin foam obtained under acidic conditions at the same aging time. This is because the phenol modified with terephthalyl alcohol under acidic conditions not only introduces the benzene ring with extremely high thermal stability into the resin structure, but also partially blocks the phenolic hydroxyl group, which is not resistant to aging. In summary, the addition of terephthalyl alcohol improves the aging resistance of phenolic foam.

## 5. Conclusions

Phenolic resin was structurally modified by terephthalyl alcohol in two different ways under different catalyst conditions. Moreover, the foaming material with the modified phenolic resin as the foaming matrix improved heat resistance and aging resistance due to the introduction of the benzene ring. The modified phenolic resin foam obtained, especially under acidic conditions, exhibited superior performance in various aspects due to its special molecular structure. The decomposition temperature at 5% weight loss was higher than that of the phenolic foam, and the temperature at which rapid weight loss occurred was also higher than that of foam. After 24 h of aging, the mass loss rate was as low as 4.5%, the pulverization rate only increased by 3.2% and water absorption rate only increased by 0.77%, the compressive strength decreased by only 20%. The dimensional change rate of the modified phenolic foam was smaller than that of the phenolic foam under the condition of prolonged heating, and the color change was slow.

## Figures and Tables

**Figure 1 polymers-11-01267-f001:**
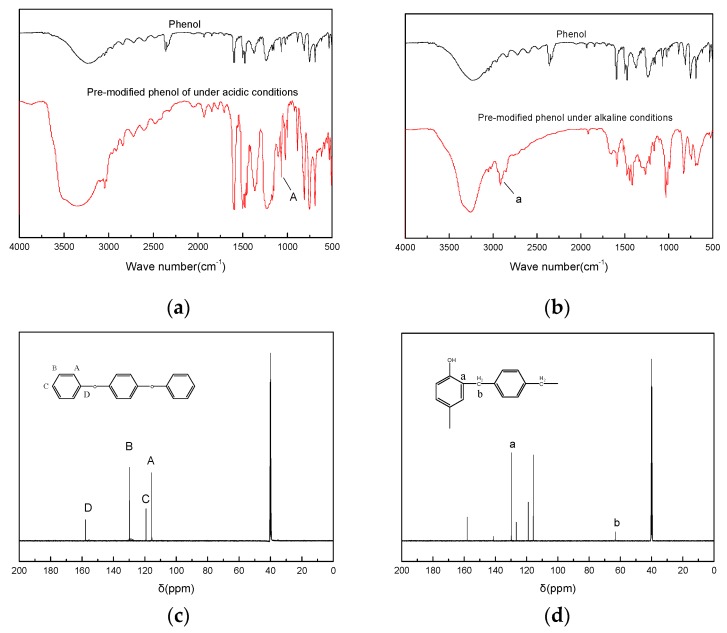
(**a**) FTIR spectra of terephthalyl-alcohol-modified phenol under acidic conditions; (**b**) FTIR spectra of terephthalyl-alcohol-modified phenol under basic conditions; (**c**) ^13^C HMR spectrum terephthalyl-alcohol-modified phenol under acidic conditions; (**d**) ^13^C HMR spectrum of terephthalyl-alcohol-modified phenol under basic conditions.

**Figure 2 polymers-11-01267-f002:**
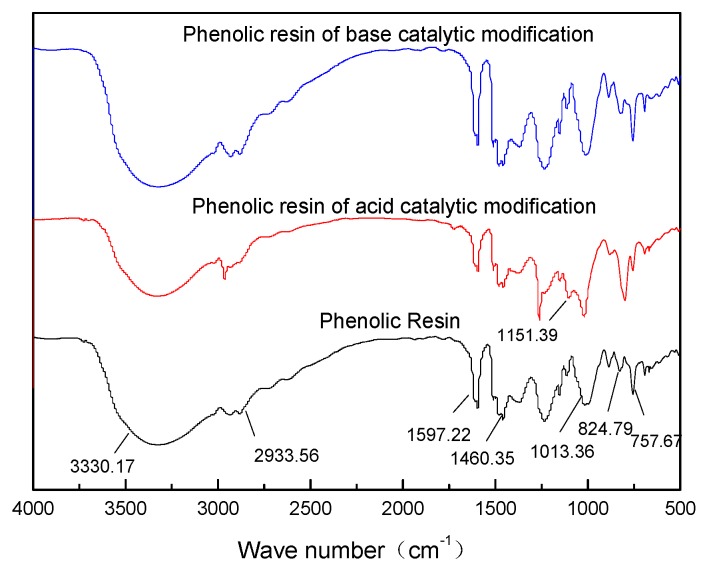
FTIR spectrum of phenolic resin and modified phenolic resin.

**Figure 3 polymers-11-01267-f003:**
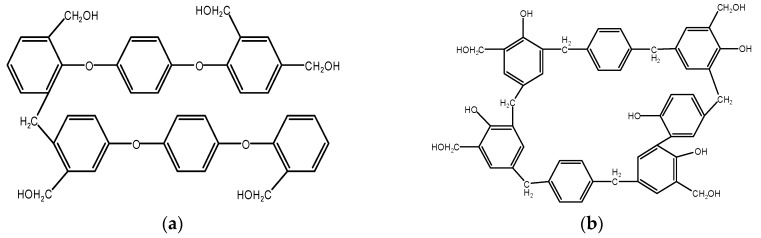
(**a**) Acid catalytic modification of phenolic resin; (**b**) base catalytic modification of phenolic resin.

**Figure 4 polymers-11-01267-f004:**
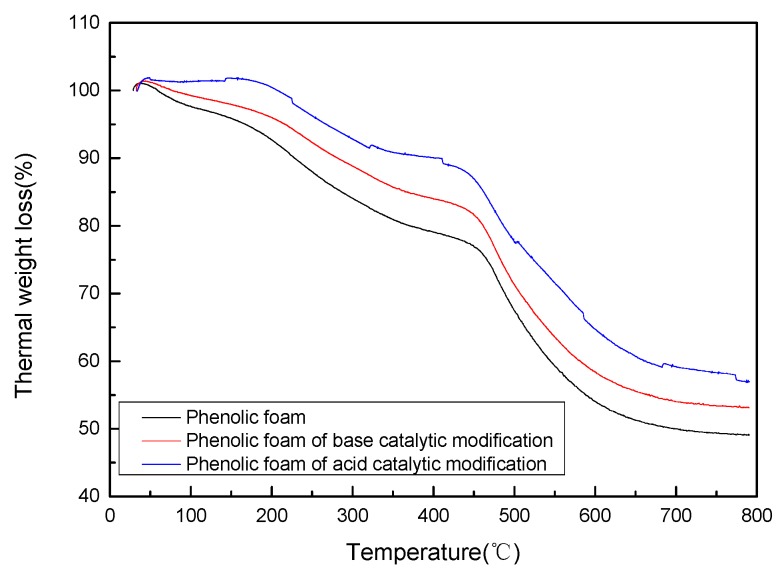
Thermogravimetric curve of phenolic foam and modified phenolic foam.

**Figure 5 polymers-11-01267-f005:**
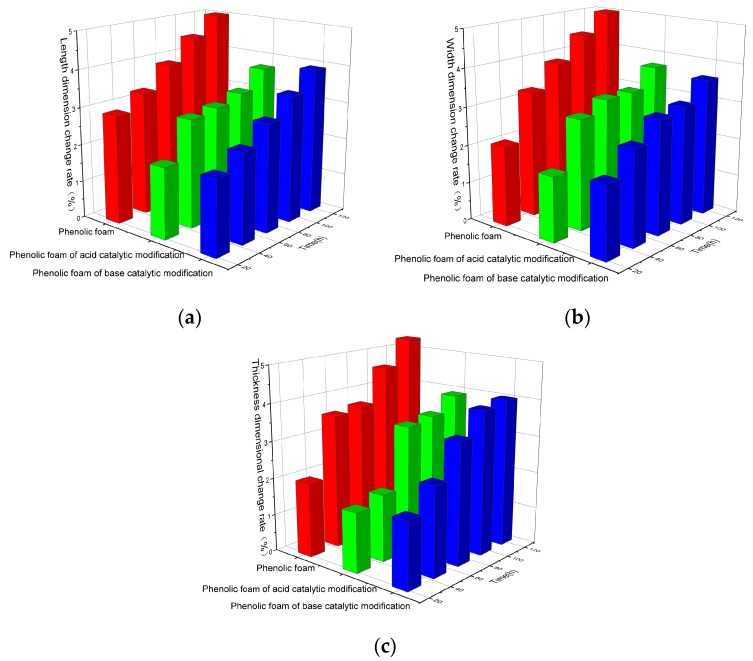
(**a**) Length dimension change rate of phenolic foam and modified phenolic foam; (**b**) width dimension change rate of phenolic foam and modified phenolic foam; (**c**) thickness dimension change rate of phenolic foam and modified phenolic foam.

**Figure 6 polymers-11-01267-f006:**
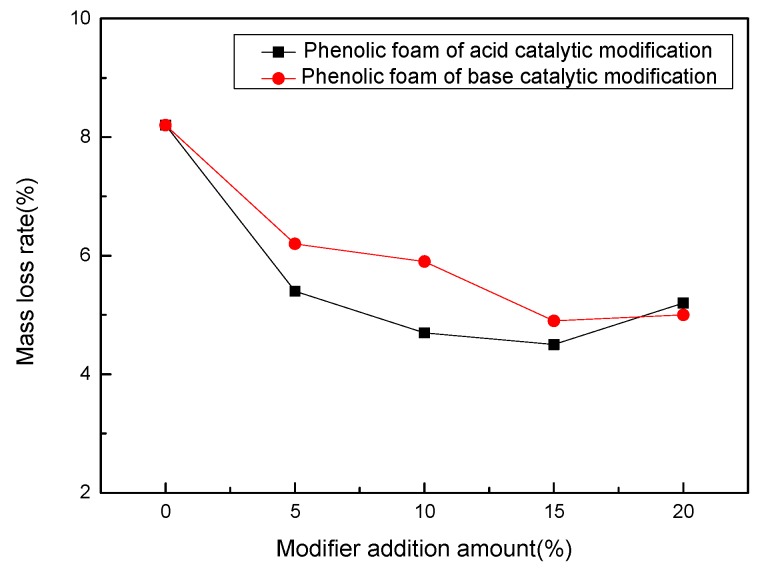
Mass loss rate variation curve of phenolic foam and modified phenolic foam.

**Figure 7 polymers-11-01267-f007:**
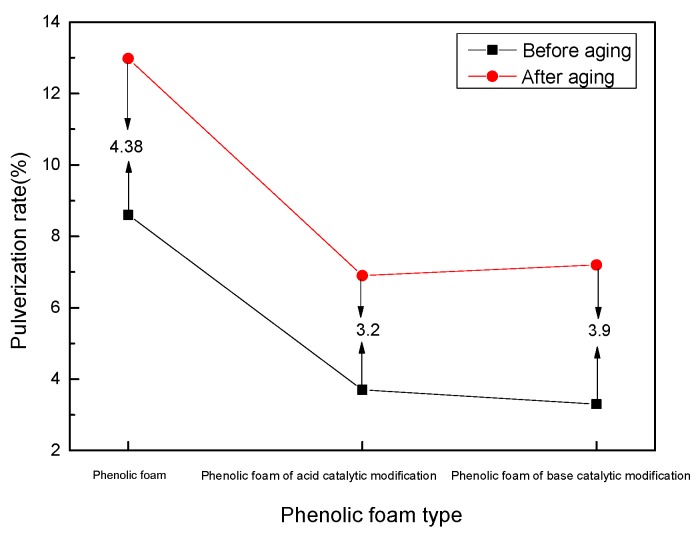
Pulverization rate variation curve of phenolic foam and modified phenolic foam before and after aging.

**Figure 8 polymers-11-01267-f008:**
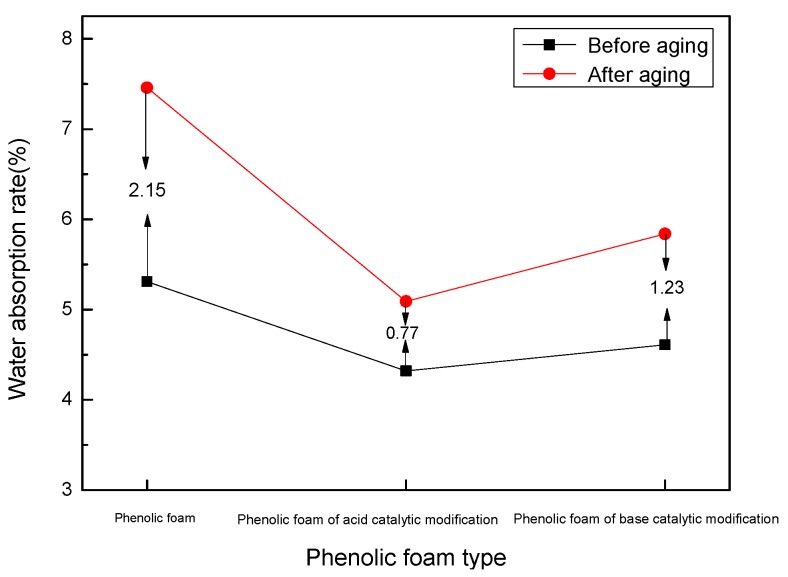
Water absorption rate variation curve of phenolic foam and modified phenolic foam before and after aging.

**Figure 9 polymers-11-01267-f009:**
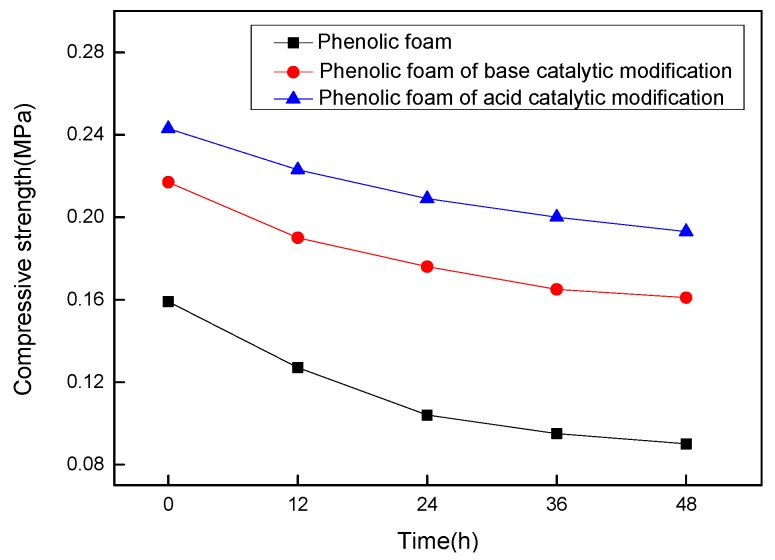
Compressive strength variation curve of phenolic foam and modified phenolic foam at certain aging time.

**Table 1 polymers-11-01267-t001:** Modified phenolic foam formula.

Modified Phenolic Resin (phr)	Tween-80 (phr)	N-Pentane (phr)	P-Toluenesulfonic Acid (phr)
100	4	12	18

**Table 2 polymers-11-01267-t002:** Basic properties of resin.

Modified Phenolic Resin	Viscosity (Pa·s)	Water Content (%)	Free Phenol (%)
Phenolic resin for foaming	3~5	6~8	<5
Phenolic resin of acid catalytic modification	4.78	7.39	3.35
Phenolic resin of base catalytic modification	4.21	7.86	3.74

**Table 3 polymers-11-01267-t003:** Thermogravimetric data analysis of phenolic resin and modified neophenol resin.

	Phenolic Foam	Phenolic Foam of Base Catalytic Modification	Phenolic Foam of Acid Catalytic Modification
Temperature 5% Weight Loss (°C)	161.76	216.36	267.58
Rapid Weight Loss Temperature (°C)	475.34	479.86	481.54
Residual Carbon Rate at 800 (°C)	49.06	53.13	56.98

**Table 4 polymers-11-01267-t004:** Schematic diagram of heat aging color change of modified phenolic foam.

	0 h	12 h	24 h	36 h	48 h	60 h	72 h
Phenolic foam	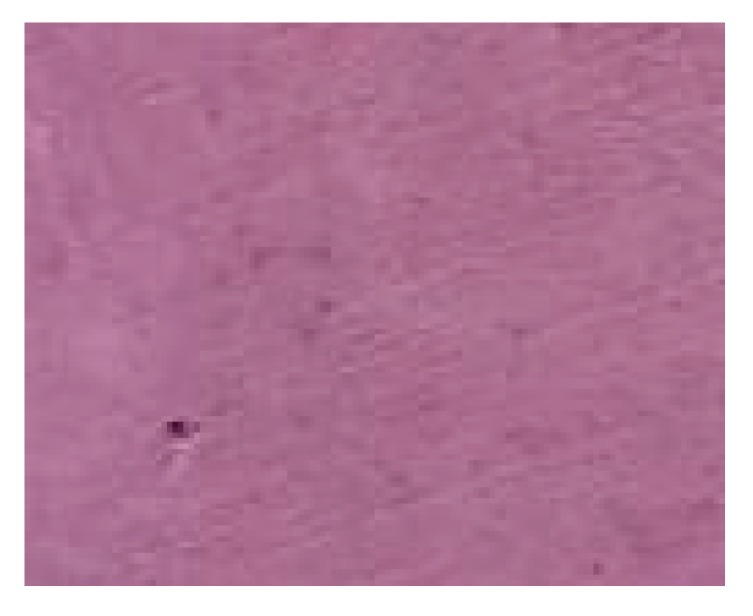	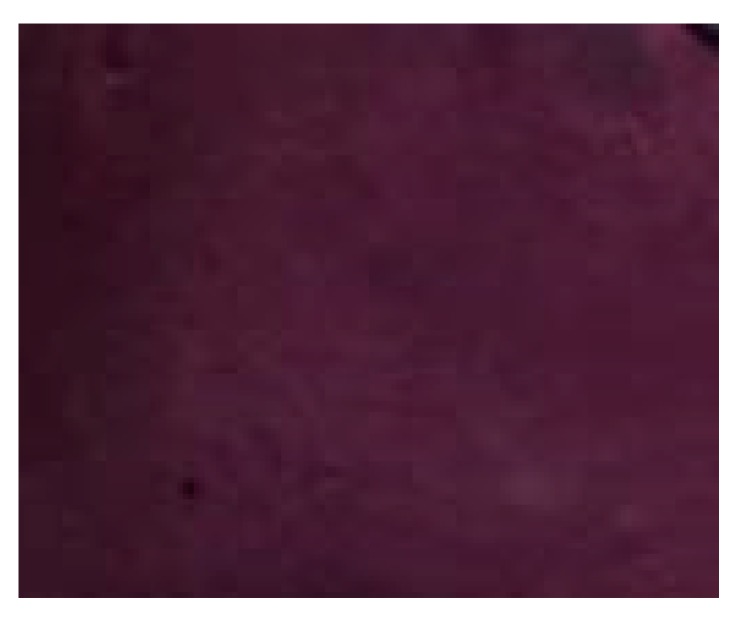	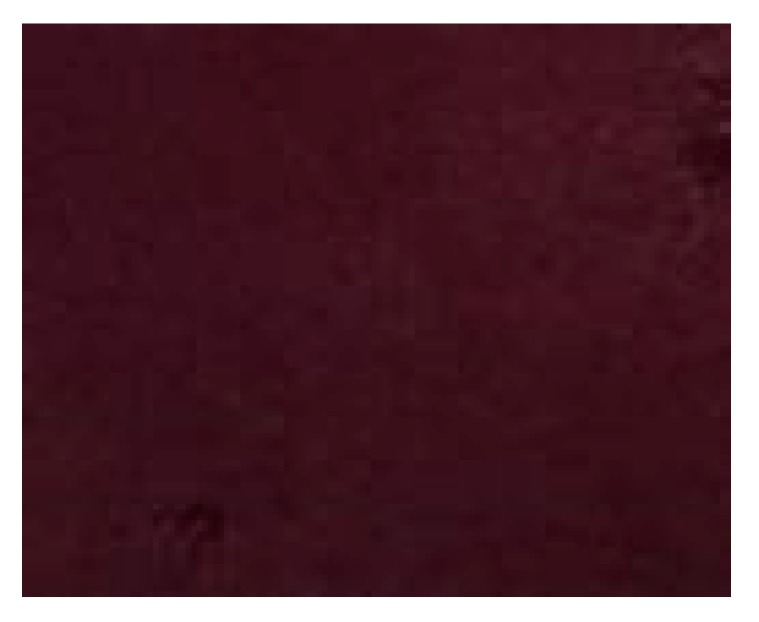	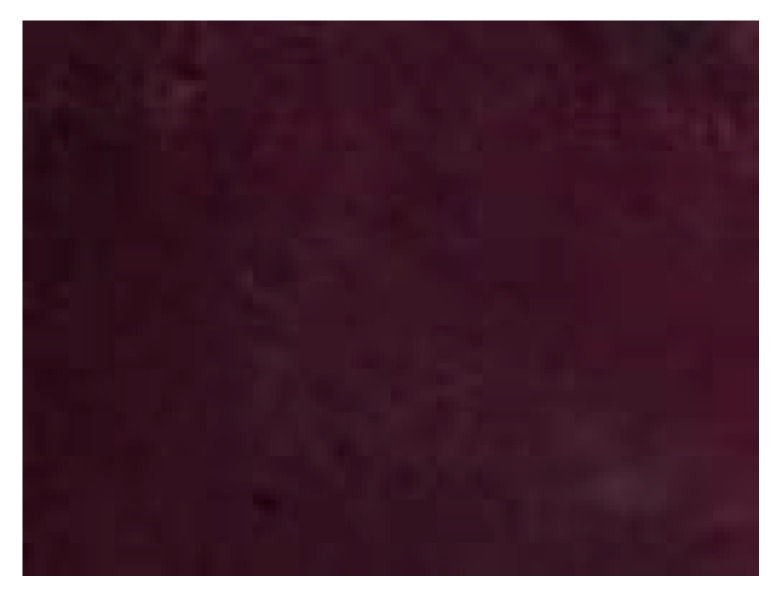	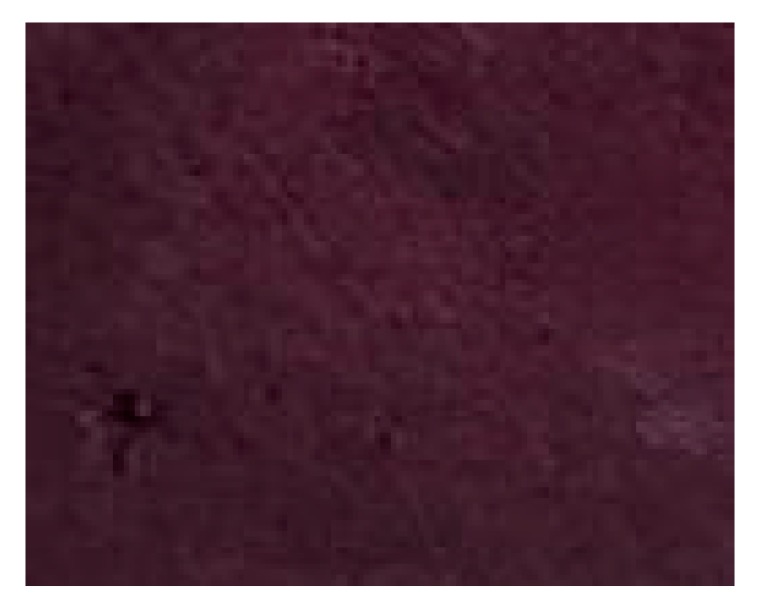	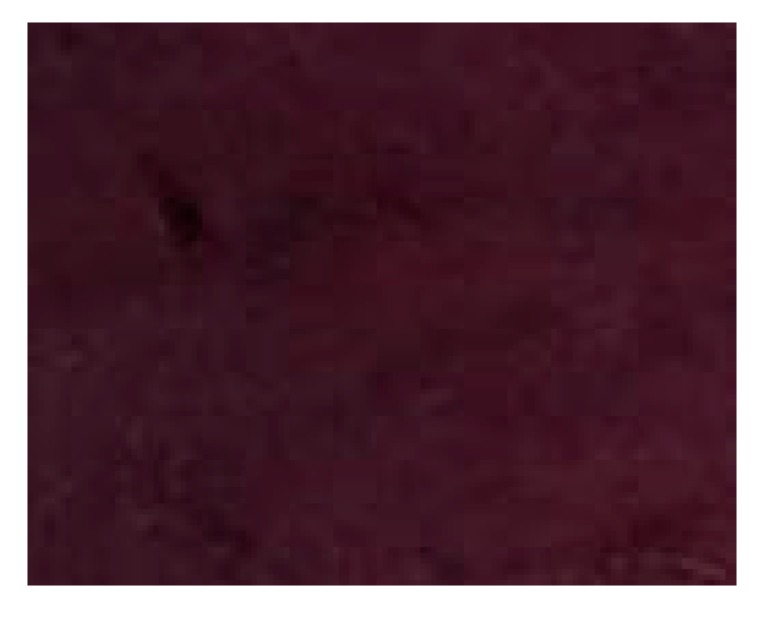	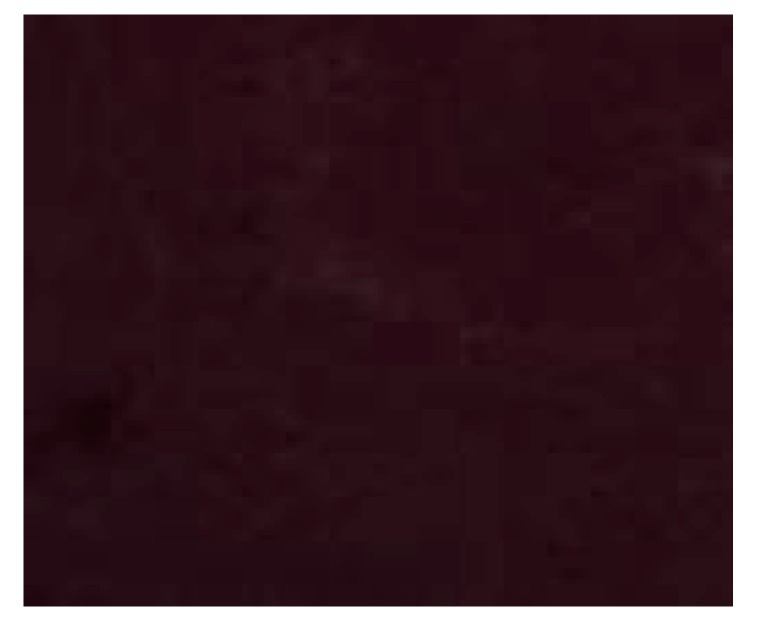
Phenolic foam of acid catalytic modification	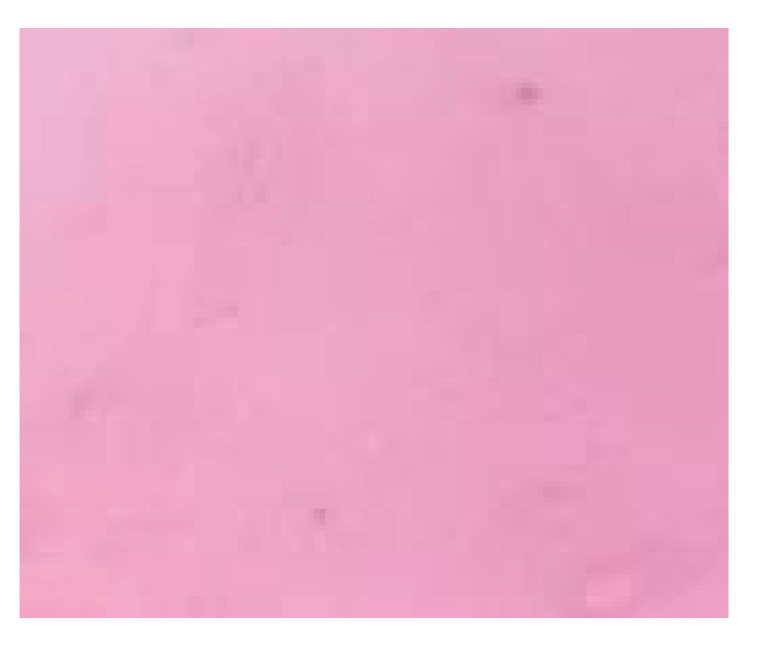	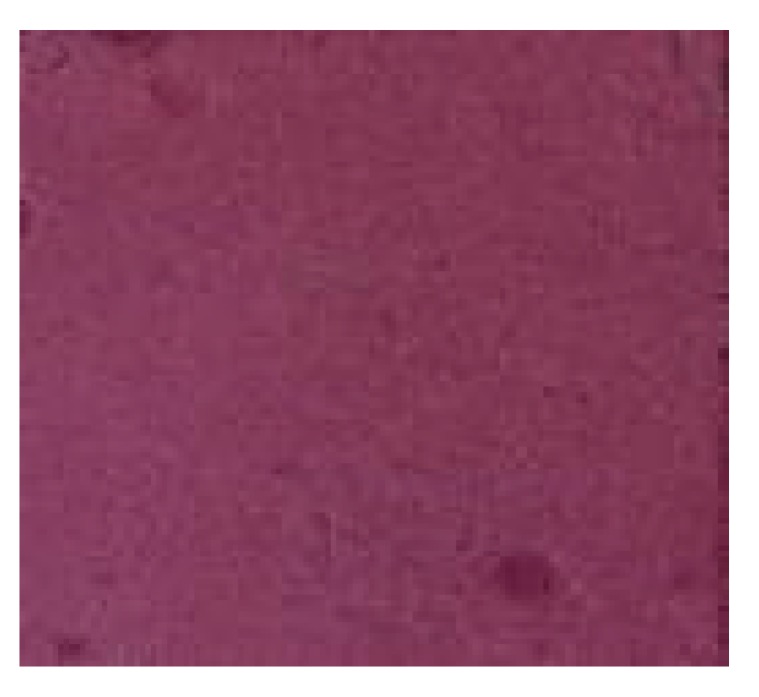	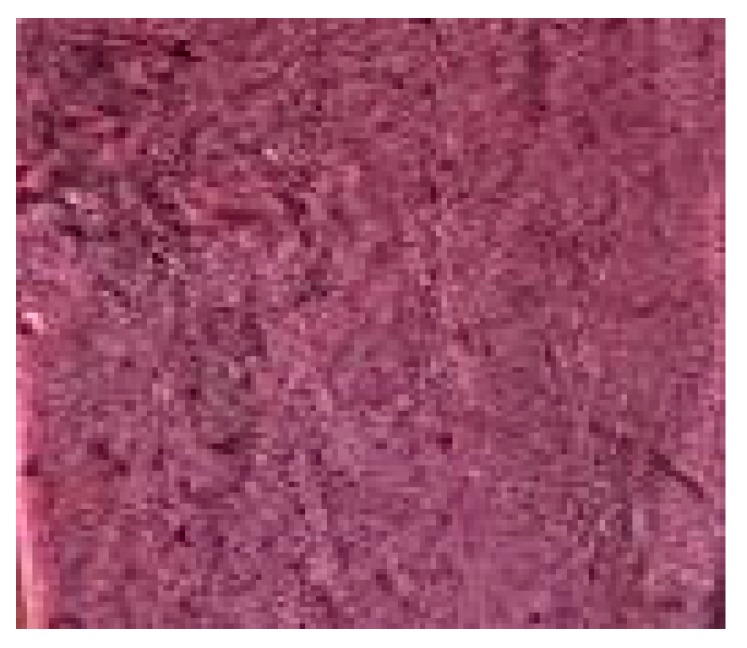	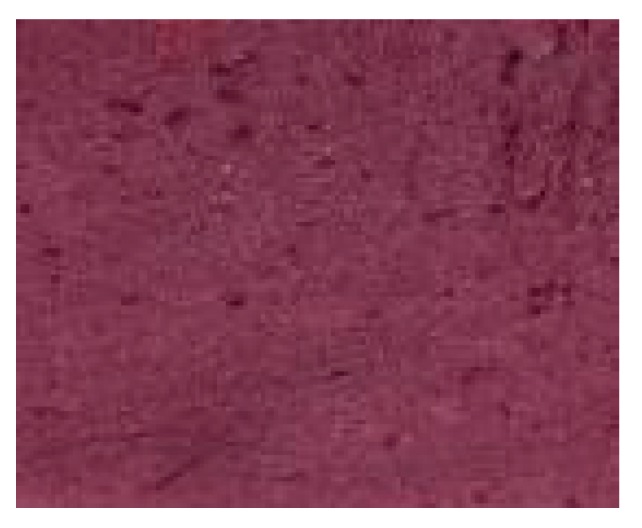	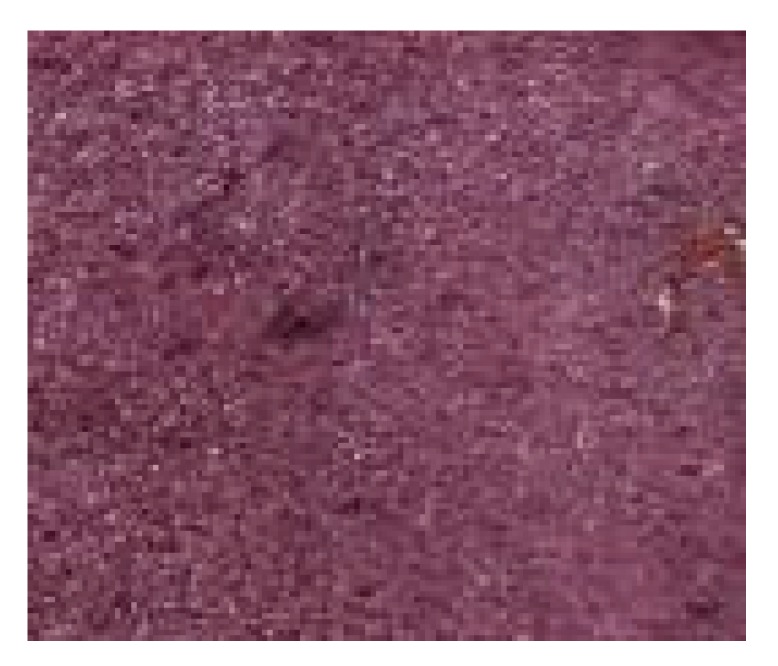	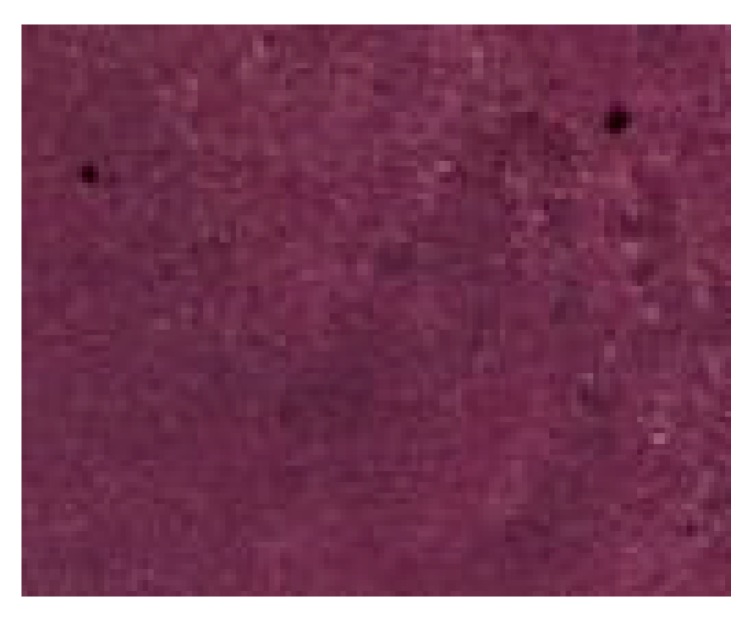	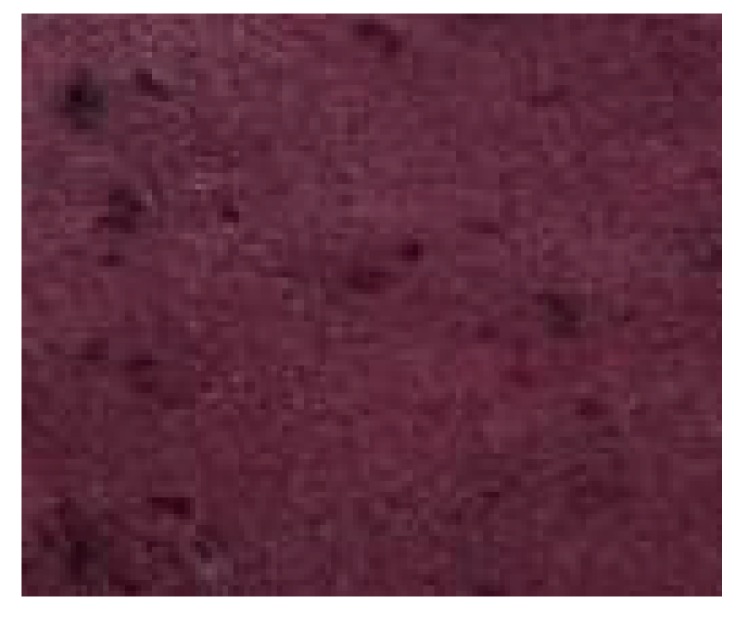
Phenolic foam of base catalytic modification	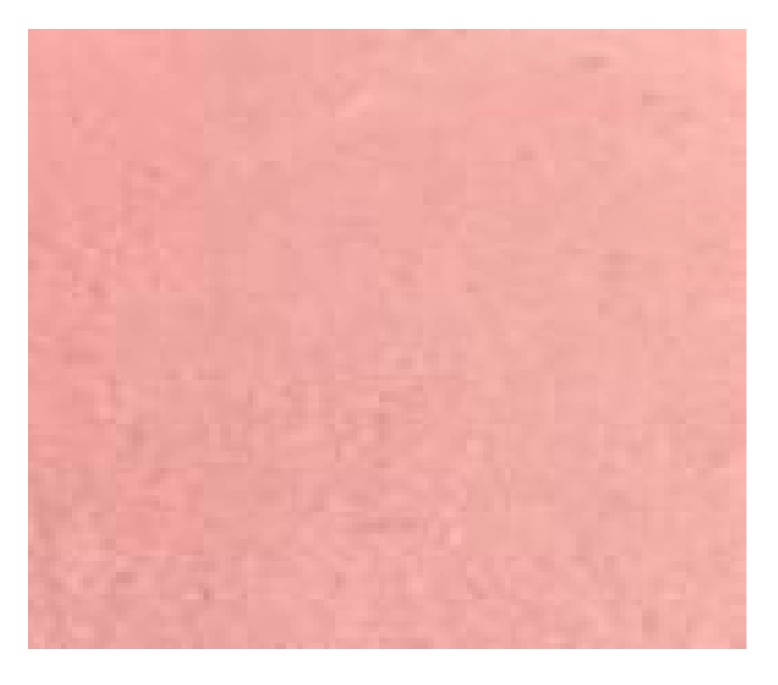	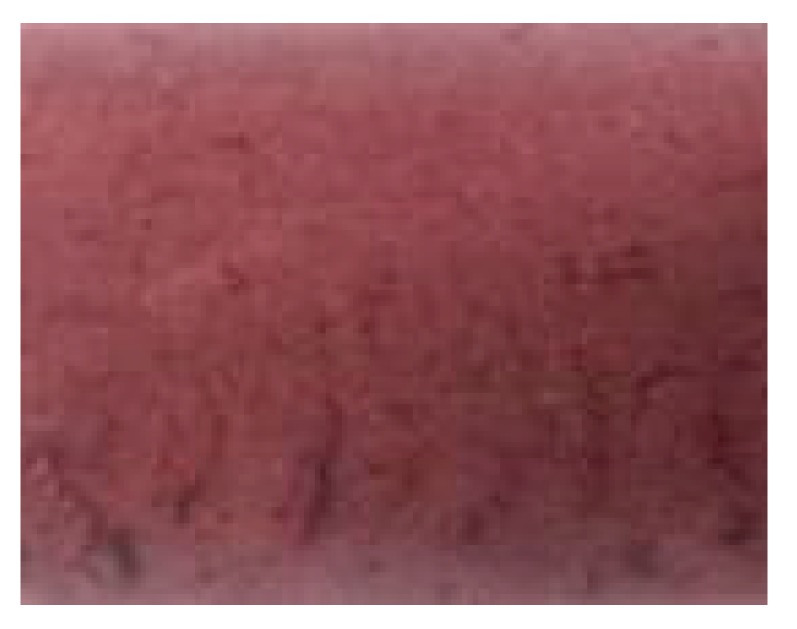	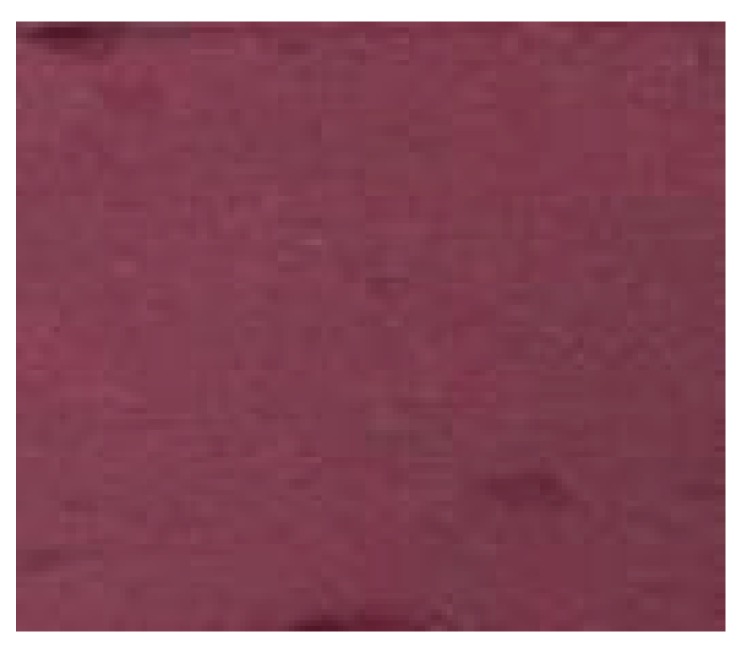	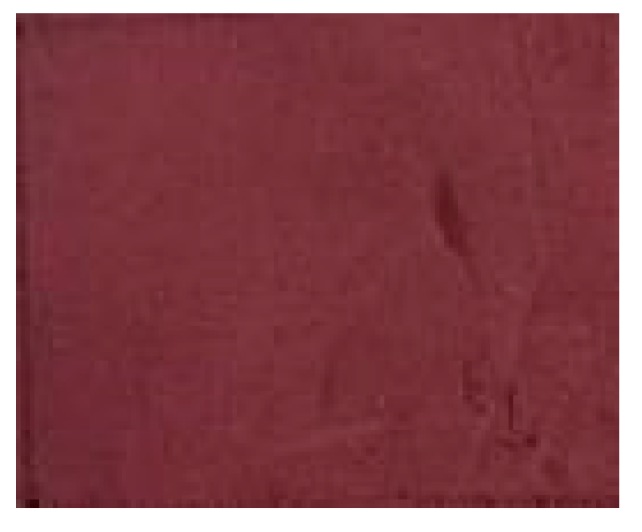	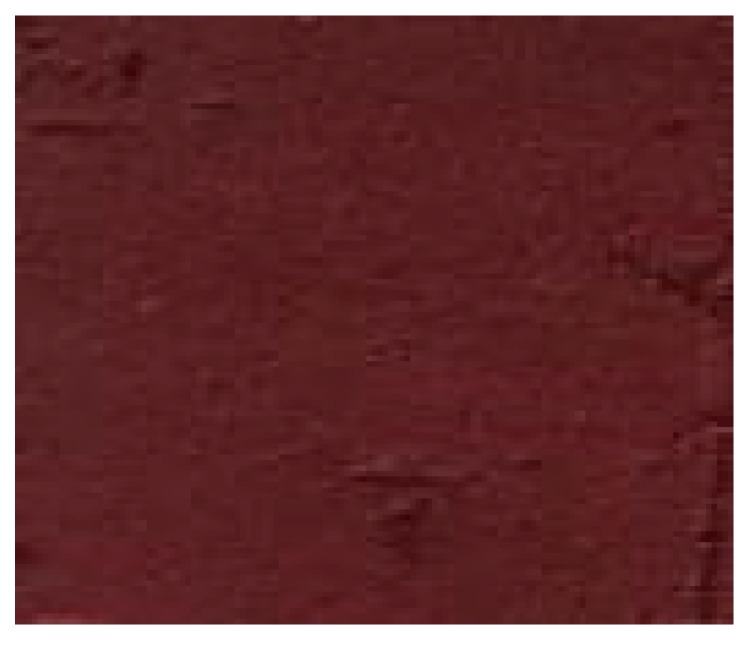	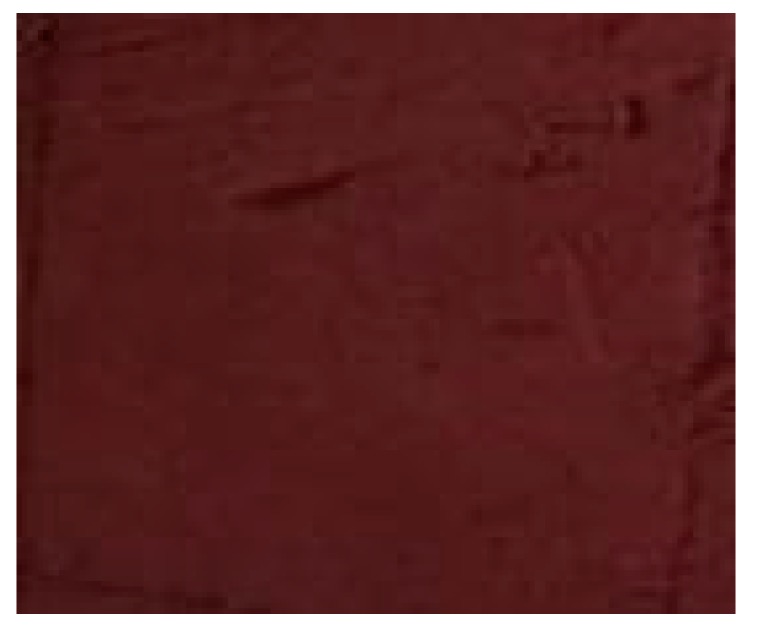	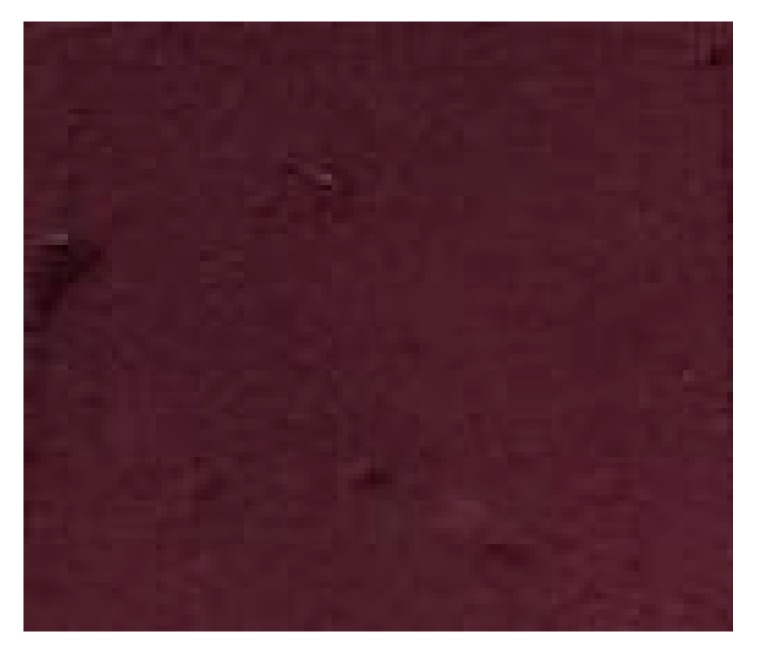

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
