# Peer review of "The Preparation and Properties of Terephthalyl-Alcohol-Modified Phenolic Foam with High Heat Aging Resistance"

_polymers, 2019, doi:10.3390/polym11081267_

Round 1
Reviewer 1 Report
A manuscript entitled "The Preparation and Properties of 3 Terephthalyl-Alcohol-Modified Phenolic Foam with High Heat 4 Aging Resistance" presents extensive research.
In my opinion Authors need to consider the following points:
In my opinion in the Section 2.4. Authors should add headers for methods used for resin characterization.
There are many editorial errors.
Author Response
Response to Reviewer 1 Comments
Dear Reviewer:
Thank you for your comments concerning our manuscript entitled “The Preparation and Properties of Terephthalyl-Alcohol-Modified Phenolic Foam with High Heat Agin Resistance”Those comments are all valuable and very helpful for revising and improving our paper , as well as the important guiding significance to our researches. We have studied comments carefully and have made correction which we hope meet with approval. Revised portion are marked in red in the paper. The main corrections and the responds are as flowing:
Point 1:In my opinion in the Section 2.4. Authors should add headers for methods used for resin characterization.
Response 1:We made the characterization as a separate part in the paper. The part 3.1 was added as a separate part of the resin characterization, and the performance test method of the resin was supplemented. The properties of the resin were added in section 4.1.2.
3.1 Characterization of resin
The modified phenolic resin was subjected to an FTIR test by Fourier transform infrared spectrometer (NEXUS 470 Thermo Electron Corporation, Shanghai, China), and the sample was evenly spread on a potassium bromide sheet and placed on an FTIR instrument for testing.
The structure of the modified phenolic resin was characterized by an NMR spectrometer (AVANCE-III-500MHz, Bruker, Switzerland). Six to eight milligrams of terephthalyl-alcohol-modified phenol product was dissolved in a solvent deuterated dimethyl sulfoxide (DMSO). The main internal standard was tetramethyl. Silane (TMS) was characterized via nuclear magnetic resonance carbon spectrum (13C NMR) operating at 125.77 MHz.
The viscosity was tested according to Chinese National Standards(GB/T 14074.3-1993).
The water content was tested according to Chinese National Standards(GB/T 14074.11-1993).
The free phenol was tested according to Chinese National Standards (GB/T 14074.13-1993).
Table 2 shows the basic properties of the modified phenolic resin, wherein the modifier is used in an amount of 15%. The viscosity of the resin used for foaming is generally controlled at 3 to 5 Pa·s, the water content is in the range of 6% to 8%, and the free phenol is controlled to be less than 5%[40]. From the table, it is known that the modified phenolic resin meets resin foaming conditions.
[40] Huang, F.; Wan, L.; Phenolic resin and its application. Chemical Industry Press 2011, 126
Point 2:There are many editorial errors.
Response 2: We had already edited the language by the MDPI platform before we submitted the manuscript. Considering your comments, we contacted the English editor of MDPI again and revised the manuscript.
Special thanks to you for your good comments and suggestions, and hope that the correction will meet your approval.

Reviewer 2 Report
The manuscript under consideration deals with preparation and analysis of properties of terephthalyl-alcohol-modified phenolic foam under acidic and under alkaline conditions. The authors state that after foaming and curing high aging resistance is achieved for modified phenolic resin. They have employed FTIR and NMR for characterization of molecular structure. The modified phenolic foam is tested for mass change rate, dimensional change rate, powdering rate, water absorption rate and compressive strength before and after aging. The authors claim that modified phenolic resin show better heat aging resistance compared to phenolic foam. The topic is interesting, however, I have some reservations regarding current version that should be addressed before consideration of this paper in “Polymers”
1. Language should be improved
2. Introduction must be improved with citing recent relevant literature and narrative must be build discussing the rational of current study
3. Authors used name of several authors and put et al at the end. It should be surname of first author et al. for example line 47, 50, 52
4. Line 64, what is polyethylene glycol diol (PEA), polyethylene glycol means it’s a diol, what is PEA, never saw this acronym/abbreviation for polyethylene glycol
5. This is off course not the first modification of phenolic resin, author should discuss and compare the improvement achieved in the properties by their modification with other relevant literature.
6. Figure 1 , its not clear and text is especially not readable, improve it
7. Figure 5, its not clear and text is especially not readable, improve it
8. Most important point, is the justification of current study and its comparison with other recent literature reports for improvement of the properties.
9. Abstract and conclusion look similar, abstract should be summary of what has been done without much on the quantitative results, that must be included in the conclusion
Author Response
Response to Reviewer 2 Comments
Dear Reviewer:
Thank you for your comments concerning our manuscript entitled “The Preparation and Properties of Terephthalyl-Alcohol-Modified Phenolic Foam with High Heat Agin Resistance”Those comments are all valuable and very helpful for revising and improving our paper , as well as the important guiding significance to our researches. We have studied comments carefully and have made correction which we hope meet with approval. Revised portion are marked in red in the paper. The main corrections and the responds are as flowing:
Point 1: Language should be improved.
Response 1: We had already edited the language by the MDPI platform before we submitted the manuscript. Considering your comments, we contacted the English editor of MDPI again and revised the manuscript.
Point 2: Introduction must be improved with citing recent relevant literature and narrative must be build discussing the rational of current study.
Response 2: Thank you for your suggestions on the introduction, we have supplemented the relevant literature in recent years based on your suggestion, and the rationality of the study is described in the last paragraph of the introduction.
Hui Liu et al. prepared a thermoset 4,4'-biphenol modified phenolic resin with a high residual carbon ratio and high thermal stability by using 4,4′-biphenol, phenol, alkylphenol and formaldehyde [38]. In order to improve the thermomechanical properties and thermal stability, Zixuan Lei used epoxy-polyhedral oligomeric silsesquioxane-modified novolac phenolic resin, and then prepared a novel organic–inorganic hybrid network with hexamethylenetetramine [39]. The phenol structure contains an active site to which terephthalyl alcohol can react with and attach the aralkyl group to the phenol structure. After the modified phenol with a different molecular structure is obtained, a modified phenolic foam with excellent heat aging resistance was prepared using paraformaldehyde.
Point 3:Authors used name of several authors and put et al at the end. It should be surname of first author et al. for example line 47, 50, 52.
Response 3:We have modified the names of quoted author in lines 47, 50, and 52 in the correct format, and other similarities have been modified.
Phosphate-containing cardanol-modified phenolic resin, prepared by Caiying Bo, et al. can overcome the brittleness of the foam, improve its flame retardancy, and attain a high thermal stability and good mechanical properties [18]. In addition, the phenolic resin may be physically modified [19,20]; that is, the modifier is blended with the phenolic resin and then foamed as a matrix resin. Yufeng Ma, et al. added an environmentally friendly, halogen-free flame retardant, a synergist, and other additives to the phenolic resin and foam, and the obtained modified foam had excellent flame retardancy [21]. Xingming Hu, et al. added polyethylene glycol (PEG) with different molecular weights and different parts to phenol-urea-formaldehyde foam.
Point 4:Line 64, what is polyethylene glycol diol (PEA), polyethylene glycol means it’s a diol, what is PEA, never saw this acronym/abbreviation for polyethylene glycol.
Response 4:“Polyethylene glycol diol (PEA)” is a translation error caused by inaccurate translation. We re-queried the correct name, changed the polyethylene glycol diol (PEA) to polyethylene adipate glycol (PEA). "polyethylene adipate glycol" can be found in the article "Influence of the hard-segment content on the shape-memory performance of thermal stimulated shape-memory polyurethanes". The following is a summary of this article:
A series of segmented thermally stimulated shape-memory polyurethanes, which consist of polyethylene adipate glycol (PEA)as the soft segment and 4,4'-diphenylmethane diisocyanate(MDI)-1,4-butanediol(BDO)-DPA(which consists of ring components) as the hard segment were synthesized by one-step polymerization. The shape recovering rates on repeated deformation of the PUs with different hard segment contents were examined. In addition, the thermal properities of the PUs were analysed by using DSC and the micro morphology was observed by using WAXD.
Point 5:This is off course not the first modification of phenolic resin, author should discuss and compare the improvement achieved in the properties by their modification with other relevant literature.
Response 5:Based on your suggestion, we have reviewed more literature on modified phenolic resins and cited them in the introduction. In section 4.1.2 of the paper, the basic properties of the modified resin and its comparison with the phenolic resin for foaming are supplemented. But the paper mainly studies the performance of phenolic foam, so there was little research on the performance of phenolic resin. We hope to get your understanding. Thank you again for your valuable suggestions.
Many research scholars have studied the modification of phenolic resins. For example, Yajun Guo et al. used nanoSiO2 to modify phenolic resin in order to enhance thermal stability [10]. Shimin Kang et al. used Humins to synthesize Humin-Phenol-Formaldehyde [11]. Yu Hu et al. used epoxy-and methacrylate-functionalized silica sols modified phenolic resin to improve the ablation resistance of their glass fiber-feinforced composites [12]. Zeya Li et al. synthesized iron-modified phenolic resin by grafting iron ions into a phenolic resin structure via a coordination reaction[13]. Shengyong You et al. synthesized KH560 modified phenolic resin by chemical synthesis using the silane coupling agent KH560[14]
[10] Guo, Y.; Hu, L.; Jia, P. Enhancement of thermal stability and chemical reactivity of phenolic resin ameliorated by nanoSiO2. Korean Journal of Chemical Engineering 2018, 35, 298-302. [10.1007/s11814-017-0240-9 ]
[11] Kang,S.; Fu, J.; Zhang, G.; Zhang, W.; Yin, H.; Xu, Y. Synthesis of humin-phenol-formaldehyde adhesive. Polymers 2017, 9, 373. [10.3390/polym9080373]
[12] Hu, Y.; Geng, W.; You, H.; Wang, Y.; Loy, D. Modification of a phenolic resin with epoxy- and methacrylate-functionalized silica sols to improve the ablation resistance of their glass fiber-reinforced composites. Polymers 2014, 6, 105-113. [10.3390/polym6010105]
[13]Li, Z.; Wu, L.; Tong, K.; Zhang, Q.; Liu, Y.; JZhang, J.; Liu,J.; Shu, S.; Xu, J.; Hu,Y. Study on the preparation of carbon nanotubes by iron modified phenolic resin pyrolysis. ,Materials Science and Engineering 2018, 423. [10.1088/1757-899X/423/1/012088]
[14] You,Y.; Dai, Y.; Dong, X.; Li, L.; Chen, Y.; Cao X. Synthesis of phenolic resin modified by silane coupling agent. Synthetic resin and plastic 2017, 34, 17-19
3.1 Characterization of resin
The modified phenolic resin was subjected to an FTIR test by Fourier transform infrared spectrometer (NEXUS 470 Thermo Electron Corporation, Shanghai, China), and the sample was evenly spread on a potassium bromide sheet and placed on an FTIR instrument for testing.
The structure of the modified phenolic resin was characterized by an NMR spectrometer (AVANCE-III-500MHz, Bruker, Switzerland). Six to eight milligrams of terephthalyl-alcohol-modified phenol product was dissolved in a solvent deuterated dimethyl sulfoxide (DMSO). The main internal standard was tetramethyl. Silane (TMS) was characterized via nuclear magnetic resonance carbon spectrum (13C NMR) operating at 125.77 MHz.
The viscosity was tested according to Chinese National Standards(GB/T 14074.3-1993).
The water content was tested according to Chinese National Standards(GB/T 14074.11-1993).
The free phenol was tested according to Chinese National Standards (GB/T 14074.13-1993).
Table 2 shows the basic properties of the modified phenolic resin, wherein the modifier is used in an amount of 15%. The viscosity of the resin used for foaming is generally controlled at 3 to 5 Pa·s, the water content is in the range of 6% to 8%, and the free phenol is controlled to be less than 5% [40]. From the table, it is known that the modified phenolic resin meets resin foaming conditions.
[40] Huang, F.; Wan, L.; Phenolic resin and its application. Chemical Industry Press 2011, 126
Point 6:Figure 1 , its not clear and text is especially not readable, improve it
Response 6: Figure 1 has been redrawn and the text portion is enlarged. In addition, we have modified all the drawings to ensure that the images are clear and readable.
Point 7:Figure 5, its not clear and text is especially not readable, improve it.
Response 7: We have redrawn Figure 5 and enlarged the text. Due to the limitations of the drawing software, the display direction of the name of the x, y, and z axes cannot be changed, but we have tried our best to make the text clear.
Point 8: Most important point, is the justification of current study and its comparison with other recent literature reports for improvement of the properties.
Response 8:After extensive review of the literature, we found that there are few articles related to phenolic foam aging research, and I am sorry that it cannot be compared with other literatures. However, in the article 4.2.1 Thermogravimetric Analysis of Modified Phenolic Foam, we supplemented the thermal properties of the foams made by other authors and the foams produced in this study.
The maximum temperatures of SGO / PF and ZGO / PF made by Xiaoyan Li et al. at 5% weight loss are 199.1°C [2]and 204°C [35], respectively, while the temperature of 5% weight loss of terephthalyl-alcohol-modified phenolic foam was as high as 267.58°C.
[2] Li, X.; Wang, Z.; Wu, L. Preparation of a silica nanospheres/graphene oxide hybrid and its application in phenolic foams with improved mechanical strengths, friability and flame retardancy. RSC ADVANCE 2015, 5, 99907-99913. [10.1039/c5ra19830e ]
[35] Satheesh Chandran, M.; Temina M.; Sunitha, K.; Dona M.; Reghunadhan Nair C. P. Allyl ether of aralkyl phenolic resin with low melt viscosity and its Alder‐ene blends with bismaleimide: synthesis, curing, and laminate properties. Polymers for Advanced Technologies 2014, 25, 881-890. [10.1002/pat.3321]
Point 9:Abstract and conclusion look similar, abstract should be summary of what has been done without much on the quantitative results, that must be included in the conclusion.
Response 9: Thank you very much for your suggestion, we have shortened the end of the summary section to avoid duplication with conclusions.
The results show that the modified phenolic foam has excellent performance. After heat aging for 24 h, the mass loss rate of the modified phenolic foam obtained by acid catalysis was as low as 4.5%, the pulverization rate was only increased by 3.2% , and the water absorption of the modified phenolic foam increased by 0.77%, which is one-third that of the phenolic foam. Compared with the phenolic foam, the modified phenolic foam shows good heat aging resistance.
Special thanks to you for your good comments and suggestions, and hope that the correction will meet your approval.

Round 2
Reviewer 2 Report
Authors have adequately addressed most of the concerns therefore i would recommend to accept this manuscript in current form